# The Perspectives of General Practitioners on the Roles and Competences of Nurses During a Euthanasia Process: A Cross-Sectional Study

**DOI:** 10.3390/healthcare13060595

**Published:** 2025-03-08

**Authors:** Dennis Demedts, Rosalie-Marianne Mathé, Johan Bilsen

**Affiliations:** Mental Health and Wellbeing Research Group, Vrije Universiteit Brussel, 1090 Jette, Belgiumjohan.bilsen@vub.be (J.B.)

**Keywords:** attitudes, competences, cross-sectional study, euthanasia, general practitioners (GPs), nursing, role

## Abstract

**Background:** The legalization of euthanasia in Belgium in 2002 positioned nurses as possible participants in the euthanasia process. However, ethical and legal dilemmas necessitate clear definitions of their roles and competencies in this context. Given that general practitioners (GPs) are the primary medical professionals performing euthanasia, understanding their perspectives on the roles of nurses is crucial. **Aim:** This study aims to describe GPs’ perceptions of the roles and competencies of nurses during the euthanasia process. The research question addressed is as follows: “What are GPs’ views on the roles and competencies of nurses in euthanasia?”. **Methods:** A quantitative cross-sectional survey was conducted among licensed GPs in Flanders, Belgium. Data were collected via an online survey using Qualtrics and analyzed with SPSS Statistics. Ethical approval was obtained from the Medical Ethics Committee of the university. **Results:** The study included responses from 237 GPs. GPs value the roles and competencies of nurses, recognizing euthanasia as a multidisciplinary process. They acknowledge the critical support nurses provide to patients and their families. GPs assert that the administration of euthanasia medication should remain the exclusive responsibility of the GP. The analysis further indicated that demographic factors did not significantly influence the results. **Discussion/Conclusions:** GPs in Belgium generally rated the roles and competencies of nurses positively. Enhancing multidisciplinary collaboration, legislating the roles of nurses, and incorporating practical euthanasia training in nursing curricula, e.g., via simulations, are recommended.

## 1. Introduction

Euthanasia is a significant and widely debated topic within the realm of ethical dilemmas globally. Euthanasia can be defined as an act whereby a person other than the individual concerned intentionally and deliberately ends the life of the individual at their explicit request [1]. Several countries have legalized euthanasia, including the Netherlands (2002), Belgium (2002), Luxembourg (2009), Colombia (2015), Canada (2016), all Australian states (2018–2023), New Zealand (2021), and Spain (2021) [2]. According to Belgian legislation, a person is eligible for euthanasia when they are in a medically hopeless condition, characterized by persistent and unbearable physical or mental suffering. It is not required for the person to be in the terminal phase of an illness [1].

All euthanasia acts must be registered with the Belgian Federal Control and Evaluation Commission for Euthanasia (FCEE) within four days after the act. In 2023, the FCEE received 3423 registration documents for euthanasia, representing a 15% increase compared to the previous year [3,4]. Most registered documents were in Dutch (70.8%, n = 2422), reflecting the more liberal Flanders. French made up a smaller proportion (29.2%, n = 1001), corresponding to the French-speaking community in mainly Wallonia and Brussels, as well as the smaller German-speaking community in eastern Belgium. About half of the euthanasia acts that took place were at home (48.6%; n = 1664), and 17.6% (n = 602) took place in nursing homes. The percentage of euthanasia cases in hospitals and palliative care units was 32% (n = 1096). Additionally, 1.8% (n = 62) of euthanasia cases occurred at other locations.

Although a euthanasia process typically involves collaboration among various healthcare providers, including nurses, the Belgian legislation primarily emphasizes the role and responsibilities of physicians. This means that the legislation mainly outlines specific expectations and responsibilities for physicians as they are the ones who make the final decision and prescribe and administer the necessary medication to perform euthanasia. While emphasizing the role of physicians is important, as they bear the ultimate responsibility and face prosecution in the case of doubt, it is equally crucial not to overlook nurses. Research shows that nurses often have a trust-based relationship with the patient, enabling them to support the patient and their relatives throughout each phase of euthanasia (clarification of the request, preparation, the act, and aftercare) [5]. So, nurses have significant added value within the overall process. Nevertheless, Belgian legislation only stipulates the role of nurses in one sentence, stating that if a nursing team is regularly in contact with the patient, the physician is advised to discuss the request with that team [1]. In contrast to Belgian legislation, in 2016, Canadian legislation recognized the importance of nurses’ roles by considering their contribution essential for enhancing communication with patients and their relatives, coordinating the process and healthcare staff, and providing support to both patients and their relatives during the process and after death [6].

Data from 2023 show that 62.6% (n = 2142) of euthanasia cases in Belgium were performed by general practitioners (GPs), often assisted by nurses [3]. Given the frequent collaboration between these professional groups, along with the legal constraints and ethical dilemmas, this study aims to provide insights into the expectations that GPs have regarding the roles and competencies of nurses during the euthanasia process.

In many countries, the roles of nurses in euthanasia remain unclear and legally undefined. While physicians bear legal responsibility, nurses often play a crucial supportive role. The level of autonomy granted to nurses varies by country—some allow independent decision-making, while others require physician supervision. This study focuses on the perspectives of GPs in Flanders, where euthanasia is legal, but the specific roles of nurses remain open to interpretation in practice. To the best of our knowledge, this is the first study to specifically analyze the perspectives of GPs regarding the nursing role in euthanasia.

## 2. Materials and Methods

### 2.1. Study Design

This study employs a cross-sectional research design to describe the perspectives of general practitioners (GPs) in Flanders (Belgium) on the roles and competencies of nurses in the euthanasia process using an online survey. Flanders was chosen as the focus because most euthanasia cases were registered in Dutch.

### 2.2. Study Population

In 2023, there were 10,332 GPs in Flanders [7]. A sample size of a minimum of 371 GPs was determined using the SurveyMonkey sample size calculator, with a confidence level of 95% and a margin of error of 5% (SurveyMonkey, San Mateo, CA, USA; Momentive, Niskayuna, NY, USA, 2024).

### 2.3. Selection and Recruitment

To include GPs in Flanders in this study, collaboration was established with HuisArtsenKring (HAK; General Practitioners’ Association, Brussels, Belgium), associations where GPs can voluntarily become members, organized based on geographical areas in Flanders, ranging from one to multiple municipalities. There are 61 HAK active in Flanders [8]. Each HAK was approached via the general email address listed on an online contact form. A request was made to forward the structured survey to all individual GPs within their circle. If no response was received from the HAK via email, follow-up phone calls were made by the researcher to discuss the request. All registered GPs who understand Dutch were invited to participate. Inclusion criteria were (1) licensed GPs practicing in Flanders, (2) the ability to understand and complete the survey in Dutch, and (3) voluntary consent to participate. Exclusion criteria included GPs who were retired or not currently practicing in Belgium.

### 2.4. Data Collection and Analysis

Data were collected using an adapted version of a questionnaire used by Demedts et al. [2,9]. The questionnaire was adapted to reflect the opinions of GPs. The survey included four demographic questions: age, gender, years of practice, and involvement in euthanasia. Additionally, the questionnaire contained 21 questions about the roles and competencies of nurses. The questions were answered using a 5-point Likert scale with the following options: “strongly disagree”, “disagree”, “neutral”, “agree”, and “strongly agree”.

The survey was created using the online questionnaire tool Qualtrics (Qualtrics; Qualtrics XM, Seattle, WA, USA, 2024). The analysis was conducted using IBM SPSS Software, Version 29 (SPSS; IBM Corp., Armonk, NY, USA, 2024). Descriptive statistics were used. Subsequently, the average scores by age category (21–30 y, 31–40 y, 41–50 y, 51–60 y, and +60 y), gender (male/female), previous involvement in euthanasia (yes/no), and years of practice as a GP (0–10 y, 11–20 y, 21–30 y, and +30 y) were compared using an independent sample *t*-test or one-way ANOVA. Independent sample *t*-tests and one-way ANOVAs were performed to examine whether demographic characteristics (e.g., age, years of experience, and prior involvement in euthanasia) influenced perceptions of nurses’ roles and competencies. These tests were chosen based on our hypothesis that experience levels and exposure to euthanasia might lead to differing perspectives among GPs.

### 2.5. Ethical Considerations

Approval for this study was obtained from the Medical Ethics Committee of Vrije Universiteit Brussel (VUB) and University Hospital Brussels. The committee granted approval on 24 April 2024 with reference number BUN 1432024000049.

To ensure the integrity of this study and the rights and well-being of participants, strict ethical principles were followed. All data were collected anonymously via a secure online platform (Qualtrics). Participants were informed beforehand that their participation was entirely voluntary and that they could withdraw at any time without consequences. Identifiable information, such as IP addresses, was not stored. The data were securely stored on the university’s internal server for the time of the study, accessible only to the researchers. Before completing the survey, participants were shown an information letter containing all relevant details of the study. The information letter provided insights into the study’s purpose, procedures, researchers’ contact information, and participation conditions. Participants had the opportunity to ask questions if anything was unclear.

## 3. Results

### 3.1. Participants

A total of 237 GPs participated in this study, with 59% being female (n = 141) and 41% being male (n = 96). The most common groups were 31–40 years (29%, n = 68) and 21–30 years (25%, n = 58). Regarding years of service, 44% (n = 105) had 0–10 years of experience. A majority of the respondents (93%, n = 221) reported having experience with euthanasia (Table 1).

### 3.2. The Roles and Competences of Nurses in Euthanasia

Table 2 presents the responses from GPs regarding the roles and competencies of nurses. More than half of the GPs disagreed with the notion that nurses hold a subordinate role to physicians, though a significant portion remained neutral or saw them as subordinate.

A large majority of GPs recognized the importance of nurses in euthanasia, with over 80% agreeing that nurses should actively support both patients and families, particularly when they are primary caregivers. Furthermore, 70% of GPs thought that discussing the euthanasia process should not be limited to the GP and the patient alone. Conversely, 55% of respondents believed that there should be no legal obligation for GPs to consult a nurse during the euthanasia process but that it should remain an advisory role. Almost all respondents believed that when a nurse has ethical objections to euthanasia, the nurse should be able to transfer care to another colleague.

Most GPs acknowledged the value of nurses in euthanasia, particularly in assessing and monitoring patients’ conditions. A strong majority (over 80%) agreed that euthanasia should be a multidisciplinary effort, reinforcing the need for interprofessional collaboration. However, 57% of GPs believed that nurses are not necessarily better qualified than physicians to assess a euthanasia request.

According to the majority of respondents, it is the nurses’ responsibility to support both the family (66%) and the patient (65%) during the act of euthanasia. A total of 63% of respondents believed that after the patient’s death, it is also the nurses’ responsibility to guide and support the family during the grieving process.

According to 57% of GPs, the preparation of the euthanaticum does not fall within the nurses’ duties. Conversely, 59% of GPs believed that nurses could be responsible for preparing general medication and 81% for setting up the IV line so that the physician can administer the euthanaticum.

GPs were divided on whether nurses should be involved in the Federal Control and Evaluation Commission for Euthanasia, with many remaining neutral on the issue.

When examining opinions on the competencies of nurses in the euthanasia process, 28% of GPs considered the knowledge of nurses sufficient, and 33% considered it insufficient. Additionally, 89% of respondents believed that sufficient simulations should be offered in nursing education to better prepare nurses for handling euthanasia requests.

### 3.3. Roles and Competences According to Demographics

The analyses based on gender, age category, years active as a GP, or involvement in the euthanasia process show no significant effects on the results. A *p*-value of 0.139 indicates no significant difference between the average scores of men and women. Similarly, a *p*-value of 0.404 suggests that age is not an influential factor in the obtained scores, while a *p*-value of 0.242 indicates that differences between groups based on years of experience as a GP are not significant. Additionally, a *p*-value of 0.400 shows no statistically significant differences related to prior involvement in euthanasia.

## 4. Discussion

This study gathered the perspectives of general practitioners (GPs) regarding the roles and competencies of nurses in the context of euthanasia. To the best of our knowledge, this is the first study to specifically analyze the perspectives of GPs regarding the nursing role in this context. The study found that GPs believed that euthanasia requests can be directed to nurses, and most thought that a multidisciplinary approach is required for the euthanasia process. These results align with the findings of Demedts et al. [2], who examined nursing students’ perspectives on their future roles and skills related to euthanasia. Given that today’s nursing students will be the future workforce, including their expectations and educational needs is crucial for shaping future policies. According to this study, almost all GPs believed that if a nurse is unable to guide the euthanasia process because of ethical concerns, a colleague can take over this responsibility, a view that also aligns with the opinions of nursing students [2]. This perspective ties into the broader ethical debate on euthanasia, which raises fundamental questions about the value of life, human dignity, autonomy, and the specific roles of healthcare professionals, including physicians and nurses. While some argue that intentionally ending a life contradicts the sanctity of life and moral principles, others maintain that individuals have the right to make decisions about their own lives, including the timing and manner of their death [10]. Additionally, more than half of the GPs believed that consulting a nurse should remain advisory rather than mandatory. This raises ethical and practical questions about whether current Belgian legislation sufficiently acknowledges the added value of nurses in the euthanasia process. The lack of a legal obligation may also reflect concerns over professional boundaries and responsibilities. Furthermore, this study highlights that almost all participating GPs considered nurses essential for assessing the patient’s medical condition, which aligns with findings from previous research [11]. Moreover, most GPs confirmed that nurses play a crucial role in both evaluating and continuously monitoring the medical condition, a finding consistent with previous research [12]. As central figures in patient care, nurses take on multiple responsibilities, including assessing needs, developing and implementing care plans, providing direct support, collaborating with other healthcare providers, and educating patients and families and advocating their well-being throughout the care process [13]. The Belgian euthanasia law [1] also recommends consulting the nursing team if they had regular interactions with the patient, a suggestion reinforced by several guidelines regarding the potential roles of nurses in euthanasia [12,14,15]. While nurses were seen as essential in assessing patient conditions, they were not considered better qualified than GPs in handling euthanasia requests. This aligns with the physician-centered approach embedded in Belgian euthanasia law but contrasts with research indicating that nurses often have closer, long-term relationships with patients, which may provide them with unique insights.

Both GPs and nursing students believe that placing the IV line is part of the nurses’ responsibilities. The act of placing the IV is not unexpected as nurses are trained to perform this procedure regularly in various contexts, making it feel familiar and secure. Additionally, more than half of the GPs indicated that nurses are not authorized to prepare the euthanaticum. It is not surprising that GPs believe nurses should not be responsible for preparing the euthanaticum nor for administering it. While it is clear that nurses are legally prohibited from administering the medication, the legality of their involvement in preparing it is less well defined. This perception may also stem from legal structures where nurses are not granted formal decision-making power in euthanasia cases. However, in countries such as Canada, nursing involvement is legally defined, leading to clearer interprofessional collaboration. Given that the executing GP holds the ultimate responsibility for the procedure, it makes sense that they would prefer to prepare the medication themselves. The guideline by De Laat et al. [12] confirms these findings and further emphasizes that collecting the patient’s wishes regarding their desired moment, clothing choice, and music choice is also part of the nurses’ duties, as indicated by 60% of GPs in this study.

The results show that GPs believe nurses play an essential role in supporting both the patient and their family during the euthanasia process. This finding is also confirmed in the study by Cayetano-Penman et al. [16]. Both the GPs in this study and the nursing students consider this support crucial, not only during the act of euthanasia but also in the aftercare [2]. This is exemplified, among other factors, by the formation of palliative home care teams, where nurses occupy a central role. The demand for nursing care is expanding, characterized by its diversity and dependent on the specific needs of each patient. This includes responsibilities such as administering medications, conducting medical procedures, monitoring vital signs, providing emotional support, and coordinating care with other healthcare professionals [17].

The opinions of the respondents in this study were divided regarding the adequacy of nurses’ knowledge to guide the euthanasia process. Nevertheless, almost all GPs believed that simulations should be offered in nursing education to better prepare nurses for handling euthanasia requests. Psychiatric nurses and nursing students themselves also indicated that they lack practical skills and desire simulations in their education [2]. Simulation training is widely acknowledged for its capacity to bridge the gap between theoretical knowledge and practical application, enhancing students’ knowledge, empathy, confidence, and communication skills while also reducing anxiety [18,19]. However, despite its proven effectiveness and common use in nursing education, simulation training is rarely applied in the areas of euthanasia and palliative care, highlighting a significant gap in educational resources [20,21]. When developing and implementing simulation modules, it is crucial to address not only the technical and content-related components but also the behavioral aspects of student nurses. A behavioral–theoretical framework, such as the Theory of Planned Behavior, can provide valuable insights into the underlying factors influencing behavior and the skills required to effectively manage euthanasia requests [22].

This study shows that demographic factors such as gender, age, years of practice as a GP, and involvement in the euthanasia process have no significant influence on the results. The lack of significant differences might reflect the strong influence of professional norms or legal regulations on the involvement of nurses in euthanasia. So, there is a clear appreciation and demand from GPs in Flanders for collaboration with nurses in guiding and supporting the euthanasia process regardless of their background or experience.

The generalizability of this study has both positive and negative aspects. A limitation is that responses were obtained from only 237 GPs, while the predetermined sample size was 370, resulting in a low response rate. This low response rate can partly be attributed to the shortage of GPs and the associated high workload, although the result is still valuable within this context. Positively, the responses come from a group of GPs with a diverse range of demographic characteristics, such as gender, age, years of experience, and previous involvement in euthanasia processes. Another limitation is that the language of this study is limited to Dutch, restricting the study population to GPs from Flanders and not including the entire country, although most euthanasia requests are conducted in Dutch [3]. The findings of this study are specific to Belgium, where euthanasia is legal but the roles of nurses in this process are not fully defined by law. In other countries, cultural and legal differences may lead to varying perspectives on nurses’ involvement in euthanasia. Future research should explore how these perceptions differ across countries with different ethical and legal frameworks.

Future research should focus on increasing the response rate to improve the representativeness and reliability of the findings. Additionally, incorporating qualitative data through a mixed-methods approach would be valuable, providing deeper insights into the respondents’ perspectives and additional contextual information.

The first recommendation emerging from this study is to strengthen multidisciplinary collaboration. The study emphasizes the need for close cooperation between GPs and nurses in guiding and supporting patients during the euthanasia process. Previous and current studies confirm that both nurses and GPs play crucial roles in guiding euthanasia [2,5,23]. Multidisciplinary collaboration offers better quality of care and leads to a better support network for the patient and their family [24].

The second recommendation is to integrate clear legislation regarding the roles of nurses. Integrating clear legislation, as described in studies by Pesut et al. [6,25,26], as in other countries where the roles of nurses are already legally established, such as Canada, is crucial to avoid legal dilemmas. In Canada, the legal recognition of the roles of nurses in euthanasia has led to more clarity and fewer ethical conflicts. Similar legislation in Flanders would better define the responsibilities and authorities of nurses, helping to reduce ethical issues and uncertainties surrounding their roles in the euthanasia process. This would benefit both healthcare providers and patients by making the process more transparent and better regulated.

The final recommendation is to focus on the education and training of nursing students. Given the importance of practical skills and the need for simulations in nursing education, it is recommended to integrate these elements more strongly into nursing curricula. Both this study and previous studies show that GPs, psychiatric nurses, and nursing students find it necessary to acquire additional knowledge and integrate simulations into nursing education to guide euthanasia requests [2]. Although simulation training is highly effective and widely implemented in nursing education, it is infrequently used in the context of euthanasia and palliative care, revealing a notable deficiency in educational resources [20,21,27]. By improving the education and training of nurses, their competencies can be strengthened, and they can fulfill their roles in the euthanasia process with more confidence. After all, today’s nursing students will soon enter clinical practice, and their educational experiences will shape their ability to navigate euthanasia cases.

## 5. Conclusions

This cross-sectional study examined GPs’ perspectives on the roles and competencies of nurses during the euthanasia process. The findings reveal that GPs hold a positive view of nurses’ contributions, underlining the value of a multidisciplinary approach. Although GPs welcome input from nurses, they maintain that the nursing role should remain advisory. Nurses are regarded as vital in assessing the patient’s condition and supporting both the patient and their family throughout the euthanasia process. However, the administration of the euthanaticum remains the exclusive responsibility of physicians.

This study underscores the importance of enhanced education and training for nurses, with nearly all GPs advocating the inclusion of simulations and practical training in nursing curricula.

Based on the findings, three key recommendations are proposed: strengthening multidisciplinary collaboration, clarifying legislation regarding nurses’ roles in euthanasia, and improving nursing education to better equip nurses for their responsibilities in euthanasia procedures.

## Figures and Tables

**Table 1 healthcare-13-00595-t001:** Demographics of respondents (n = 237).

		n	%
Gender	Male	96	41
	Female	141	59
	Other	0	0
Age	21–30 y	58	25
	31–40 y	68	29
	41–50 y	27	11
	51–60 y	50	21
	60+ y	34	14
Years as GP	0–10 y	105	44
	11–20 y	38	16
	21–30 y	45	19
	+31 y	49	21
Ever involved in euthanasia?	Yes	221	93
	No	16	7

**Table 2 healthcare-13-00595-t002:** Nurses’ roles and competences in euthanasia (n = 237).

I Am of the Opinion That…	Totally Disagree	Rather Disagree	Neither Agree nor Disagree	Rather Agree	Totally Agree
	n (%)	n (%)	n (%)	n (%)	n (%)
The role of nurses					
(1a) nurses are in a hierarchically subordinate role to the physician.	47 (20%)	74 (31%)	50 (21%)	52 (22%)	14 (6%)
(1b) nurses should play an active role in assisting patients and their families during the euthanasia process?	4 (2%)	6 (3%)	27 (11%)	110 (46%)	90 (38%)
(1c) the primary nurse should be involved by the physician in the euthanasia request.	4 (2%)	12 (5%)	18 (8%)	102 (43%)	101 (43%)
(1d) discussing euthanasia belongs only to the GP and the patient themselves.	63 (27%)	101 (43%)	26 (11%)	28 (12%)	19 (8%)
(1e) the nurse should assess the patient’s condition.	3 (1%)	3 (1%)	19 (8%)	107 (45%)	105 (44%)
(1f) the nurse can add value to the process by closely observing and measuring the patient’s medical condition.	4 (2%)	2 (1%)	11 (5%)	73 (31%)	147 (62%)
(1g) a euthanasia request can be asked to a nurse.	19 (8%)	33 (14%)	37 (16%)	87 (37%)	61 (26%)
(1h) a euthanasia request needs a multidisciplinary approach.	5 (2%)	19 (8%)	19 (8%)	78 (33%)	116 (49%)
(1i) it is the nurse’s job to support the family when euthanasia is performed.	1 (<1%)	23 (10%)	57 (24%)	115 (49%)	41 (17%)
(1j) it is the duty of the nurse to assist the patient during the act of euthanasia.	2 (1%)	18 (8%)	64 (27%)	115 (49%)	38 (16%)
(1k) it is the nurse’s job to prepare general medication.	16 (7%)	31 (13%)	52 (22%)	101 (43%)	37 (16%)
(1l) it is the nurse’s job to prepare the euthanatic agent.	61 (26%)	74 (31%)	56 (24%)	35 (15%)	11 (5%)
(1m) it is the nurse’s job to place an IV so that the physician can administer the euthanasia drug.	6 (3%)	9 (4%)	31 (13%)	80 (34%)	111 (47%)
(1n) it is the nurse’s job to assist the patient’s family after death.	5 (2%)	15 (6%)	68 (29%)	113 (48%)	36 (15%)
(1o) it is the duty of nurses to enquire about the practical wishes of the patient and their relatives (e.g., time of day, choice of clothing, possible choice of music, etc.).	4 (2%)	31 (13%)	58 (24%)	110 (46%)	34 (14%)
(1p) that there should be a legal regulation to oblige GPs (instead of advising) to consult a nurse in the process of euthanasia.	59 (25%)	71 (30%)	54 (23%)	32 (14%)	21 (9%)
(1q) a nurse, who has ethical concerns about euthanasia, should be able to transfer care to another colleague.	2 (1%)	3 (1%)	5 (2%)	50 (21%)	177 (75%)
(1r) nurses know the patient better and therefore they are better able to assess the request for euthanasia than doctors.	46 (19%)	90 (38%)	67 (28%)	27 (11%)	7 (3%)
(1s) nurses should be part of the Federal Euthanasia Control and Evaluation Commission.	22 (9%)	23 (10%)	102 (43%)	63 (27%)	27 (11%)
The competencies of nurses					
(2a) nurses have sufficient knowledge to deal with a patient’s euthanasia request. They know what to do.	19 (8%)	60 (25%)	92 (39%)	52 (22%)	14 (6%)
(2b) there should be simulation offered in the nursing course to learn how to deal with a euthanasia request.	2 (1%)	7 (3%)	16 (7%)	103 (43%)	109 (46%)

## Data Availability

The original data presented in this study are openly available in Mendeley Data at https://doi.org/10.17632/gyxswz63r6.1 (assessed on 6 February 2025).

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
