# Peer review of "The Perspectives of General Practitioners on the Roles and Competences of Nurses During a Euthanasia Process: A Cross-Sectional Study"

_healthcare, 2025, doi:10.3390/healthcare13060595_

Round 1

Reviewer 1 Report

Comments and Suggestions for Authors

Dear authors,
The article addresses a critical area of the role of nurses because it impacts on the beliefs, values and foundations of the nursing profession. 
The introduction should be improved, showing the best problem under study. On the other hand, data is presented that is not correct (it states that Portugal legalized euthanasia in 2023, which is not true, it is a country where this practice is not legal). I would also recommend that they make it clear that the evolution of the nursing profession is different in all countries, an aspect that is not clear in the study. There are countries where nurses can develop autonomous interventions, without being interdependent on another professional.
It should be made clearer how the confidentiality and anonymity of the participants will be guaranteed, since the data will be collected online. Also, where will the data be stored and for how long?
In the results, I recommend that you don't repeat data that is in the table, just describe the most relevant data, using a percentage or numerical value.
Another important aspect that should be noted is that this study, given the subject matter, may not be generalizable to other countries, as a result of the philosophical convictions of each country.

Reviewer 2 Report

Comments and Suggestions for Authors

Dear Authors, thank you for allowing me to review this interesting manuscript describing the perspective of GPs on the role and competencies of nurses during euthanasia. 

I found the article well-written, with a clear objective and a straightforward methodology. Moreover, I found the topic very timely. I have some comments for improvements, reported below.

Abstract. It is clear, but I would suggest to specify in the Methods the design you have applied.

Introduction. It is well written and straightforward to the conduce the reader into the aim of this study. The contextual framework is adequately underpinned (with the geographical and legal aspects well focused). 

Methods/study design. Please, correct the verb at line #81 from "analyze" to "describe".

Selection and recruitment. Please, specify the inclusion and exclusion criteria.

Data collection and analysis, please link your analysis plan to your hypothesis. Why did you perform the t-test and anova?

Results. They are adequately structured and clearly written.

Discussion. In general, I would suggest to avoid the repetition of the results into the discussion (ie., line 193, 209, 219, ...). Moreover, I did not find that some central points in your results were discussed. I am referring to the perceived subordinated position of the nurse, the reported absence of legal obligation for GPs to consult a nurse, and the fact that nurses are not neccessarily better qualified than GPs.

On the other hand, you deeply focused on knowledge acquisition and simulation, that is great.

Lastly, you often refer to nursing students, but it is not always clear why.

I hope these comments will help you in improving the quality of your manuscript.

Round 2

Reviewer 1 Report

Comments and Suggestions for Authors

Dear Authors,

This article is accept in present form.

Best regards,

Comments on the Quality of English Language

The English could be improved to more clearly express the research.